# The Physiological Response of the Fiddler Crab *Austruca lactea* to Anthropogenic Low-Frequency Substrate-Borne Vibrations

**DOI:** 10.3390/biology14080962

**Published:** 2025-07-31

**Authors:** Soobin Joo, Jaemin Cho, Taewon Kim

**Affiliations:** 1Department of Ocean Sciences, Inha University, Incheon 22212, Republic of Korea; 2Program in Biomedical Science and Engineering, Inha University, Incheon 22212, Republic of Korea

**Keywords:** substrate-borne vibration, fiddler crab, anthropogenic substrate-borne vibration, biomarker

## Abstract

To investigate whether anthropogenic vibrational disturbance induces physiological stress in the semi-terrestrial crab *Austruca lactea*, an exposure experiment was conducted using a vibration signal that mimicked pile-driving operations. Under exposure to 250 Hz vibration, lactate concentration increased, while ATP concentration decreased, in the crab’s leg muscle. Additionally, the expression of heat shock protein (70 kDa) in one individual increased by 17-fold under the same condition. These findings suggest that vibrational disturbance can have negative physiological effects on marine animals, and such physiological indices may serve as biomarkers for assessing noise pollution, including vibrational disturbance.

## 1. Introduction

Anthropogenic vibrations and noise are recognized as pollutants in the marine environment, yet evaluations of their various sources and effects on different species remain insufficient. Additionally, such sources may produce various types of continuous or impulsive stimuli and increase background noise through intense disturbances [1,2]. Since the Industrial Revolution, human activities have introduced and increased noise in marine environments over time [2,3,4], and vibrational disturbances are also believed to have increased alongside underwater noise [5,6,7,8]. In particular, pile-driving is a commonly used construction method in coastal developments and is essential for installing monopile or jacket-type structures in offshore wind farms [9]. However, pile-driving generates various types of vibrations, including shear, compressional, and surface waves [5], which are emitted from both the side (skin) and tip of the pile during operation [10]. Surface waves such as Rayleigh, Love, Sholte waves, and bending waves are collectively referred to as substrate-borne vibrations, and animals that use vibrational cues are known to be particularly sensitive to these signals [11].

Hill and Wessel [12] proposed the concept of “biotremology,” which refers to the detection or generation of vibrational stimuli by animals as signals or cues to gather information from their environment. Since substrate-borne vibration serves as a crucial communication channel for many species, such stimuli provide essential information that enables animals to adapt to or respond to changing ecological conditions [13]. In marine environments, vibrational stimuli including signals and cues are particularly important for benthic organisms. Contrary to earlier assumptions that vibroacoustic detection in marine invertebrates is limited [14], recent studies have provided increasing evidence of their capacity for detecting such stimuli. In decapod crustaceans, including crabs, various sensory organs such as statocysts, sensory hairs, and chordotonal organs contribute to vibroacoustic perception [1,15]. Notably, fiddler crabs possess a uniquely developed organ known as Barth’s myochordotonal organ, and semi-terrestrial crabs including fiddler crabs have shown greater sensitivity to vibroacoustic stimuli than other crab species [14,16,17]. In addition, fiddler crabs utilize vibrational signals such as drumming for intraspecific communication during agonistic interactions and mating behaviors [18,19]. In particular, Kim et al. [18] reported that the vibrational characteristics of the drumming behavior, a known vibrational behavior of the white-clawed fiddler crab (*Austruca lactea*), differed depending on the context of communication, such as courtship or agonistic interactions. Given the importance of vibrational communication in fiddler crabs, anthropogenic vibrational disturbances may have detrimental effects on their behavior and ecology.

However, despite the known vibroacoustic detection ability of fiddler crabs, the effects of vibrational pollution on this group have rarely been documented. In particular, as *A. lactea* is listed as an endangered species in Korea, understanding these effects is important from a species conservation perspective. Joo and Kim [20] reported that behavioral responses related to locomotion were altered under exposure to substrate-borne vibrations mimicking pile-driving. Under vibration at frequency of 120 and 250 Hz, *A. lactea* exhibited decreased movement duration and increased walking velocity. Such alterations in locomotion are related to disruptions of normal behavior and increased energy consumption, suggesting the possibility that such vibrations may act as a stressor. Additionally, there is evidence that vibroacoustic disturbance affects crustaceans by disrupting sensory information use, altering behavior, and impairing antipredator responses [21,22,23]. Filiciotto et al. [24], however, emphasized the importance of integrating behavioral and physiological assessments to better understand the impacts of anthropogenic noise. Various previous studies on crustacean species have investigated physiological responses to noise pollution using bioindicators such as protein and lactate concentrations, gene expression (e.g., heat shock proteins), DNA damage, total hemocyte count, and oxygen consumption [24,25,26,27,28]. Crustaceans experience stress analogous to that of vertebrates, which can be expressed through measurable physiological responses [29]. Therefore, in this study, we investigated the physiological responses of *A. lactea* to anthropogenic substrate-borne vibrations at 120 and 250 Hz, as referenced in Joo and Kim [20], and discussed whether these responses can be used as biomarkers for assessing the effects of vibrational pollution on this species.

To examine the physiological responses of *A. lactea* to anthropogenic substrate-borne vibrations, biochemical analyses were conducted following a controlled exposure experiment. Based on previous studies, ATP and lactate concentrations in leg muscle were selected as candidate biomarkers to assess the effects of vibrational pollution [24,30,31]. Additionally, the expression of heat shock proteins (HSPs) in the soft tissue of *A. lactea* was assessed after vibration exposure [24,28]. In particular, since the HSP sequences of *A. lactea* have not been previously identified, this study includes newly designed primer sequences to amplify the HSP genes in *A. lactea*. We predicted that ATP concentrations would decrease under exposure to substrate-borne vibrations compared to the control group, while lactate concentrations and HSP70 expressions would increase. The experiment was conducted under laboratory conditions using a vibration generator to produce quantifiable vibrations, and the biochemical indices were subsequently analyzed.

## 2. Materials and Methods

### 2.1. Target Species and Incubation

The crabs were collected from the uppermost intertidal zone near Yeongjong Bridge on Yeongjong Island, Korea (37.52° N, 126.54° E; Appendix A). The crabs were collected by hand and with a shovel during the summer months of August and September 2024. All subsequent experimental procedures (acclimation, treatment, and analysis) were carried out in the laboratory at Inha University (Incheon, Republic of Korea). Only adult male crabs were with a carapace width of approximately 2 cm were collected, consistent with a previous study [20]. Immediately after collection, the crabs were transferred to the laboratory and acclimated for over one week in a custom-designed aquarium specifically constructed for housing *A. lactea*. As a semi-terrestrial species that is surface-active during low tide [32], the acclimation aquarium was not fully submerged. Instead, an outlet was installed to allow drainage, and a sump tank was placed beneath the acclimation aquarium to continuously supply seawater, ensuring that the sediment remained moist. The sediment used in the aquarium was collected from the crabs’ habitat and maintained at a depth of approximately 3 cm. To maintain appropriate salinity levels, tap water mixed with Red Sea Salt (Red Sea, Houston, TX, USA) was used to keep the sump at 32 psu. Temperature was controlled at 30 °C for both air and water using a two-way air conditioner (AC023RA1PBH1SY, Samsung Electronics Co., Ltd., Suwon, Republic of Korea). Salinity and temperature were monitored using a multi-sensor meter (YSI Pro 2030, Yellow Springs Instruments Inc., Yellow Springs, OH, USA). An external filter (EHEIM Professionel 4+ 250, EHEIM GmbH & Co. KG, Deizisau, Germany) was installed to help maintain water quality.

### 2.2. Experimental Setup and Procedures

Most of the experimental setup for vibration exposure in this study was based on the work of Roberts et al. [33], Roberts et al. [34], and Joo and Kim [20]. An electromagnetic shaker (LDS v101, BRÜEL & KJÆR, Marlborough, MA, USA), connected to an amplifier (LDS LPA100, BRÜEL & KJÆR, USA) and a function generator (AFG1062, TEKTRONIX, INC., Beaverton, OR, USA), was installed on the ceiling of a custom steel-framed experimental structure (Figure 1). The signal generated from the function generator was amplified and delivered to a steel M4 bolt mounted on the shaker, producing the desired vibration. The selected frequencies were 120 Hz and 250 Hz based on the results of Joo and Kim’s study [20], applied with a 0.8 s interval using a sine wave modulated with a square-shaped AM waveform (Figure 2). The vibration amplitude was maintained at 107.22 ± 0.62 dB re 1 µm/s^2^ (mean ± standard deviation) across all trials. The vibrational characteristics, including amplitude, temporal pattern, and frequency, were configured to mimic those generated by pile driving activity [21,35,36,37] (detailed in [20]). To minimize external interference, the experimental box containing sediment was placed on an anti-vibration pad. The box (30 × 30 × 30 cm; W × L × H) was divided by a central mesh net to prevent the crabs from contacting the vibration source. One side housed the crab, and the other contained the vibration source with the M4 bolt embedded in the sediment. The vibrations were measured using a high-sensitivity accelerometer (KS48C, Metra Mess-und Frequenztechnik, Radebeul, Germany; frequency response: 0.1–4000 Hz, measuring range: ±0.6 g, single axial) placed in the crab compartment on sediment surface. Data were collected using a data logger (DT9837, Measurement Computing Co., Newbury, UK). Prior to the experiment, the accelerometer was calibrated using the shaker. Vibration data were saved in CSV format via QuickDAQ software (Version 3.7.0.49, Measurement Computing Co., Norton, MA, USA) and analyzed using fast Fourier transformation (FFT) and short-time Fourier transformation (STFT) with the Signal Processing Toolbox 23.2 in MATLAB R2023b. The signal analysis results are presented in Figure 2 and Appendix A, confirming 120 Hz and 250 Hz as the dominant frequencies for each experiment.

To concurrently conduct the exposure experiment with a control group, two crabs were tested per trial: one allocated to the experimental box and the other to a control box designed under identical conditions, except without vibration generation. Background vibration in the control box was minimized using an anti-vibration pad and sufficient distance from the exposure box. The crabs were acclimated for over 5 min after being transferred to each box, with the surrounding area covered by blackout curtains to minimize external stimuli. Vibrations at either 120 Hz or 250 Hz were then generated using a function generator for a duration of 30 min. Each frequency experiment was repeated eight times, resulting in a total of 32 crabs tested (*N* = 8 per frequency group). After exposure, the walking leg and hepatopancreas were collected into 5 mL tubes, rapidly frozen in liquid nitrogen, and stored at −80 °C for further analysis.

### 2.3. Biochemical Measurements

Lactate and ATP concentrations were measured in the soft tissue of the walking legs. Due to the small size of the legs, the tissue volume from a single leg was insufficient, and it was difficult to extract soft tissue individually (personal observation). Therefore, all walking legs were used collectively for measurements. To efficiently extract soft tissue, the entire legs were homogenized using the Ultra AUTOMILL (UTK-AM7, Tokken, Inc., Chiba, Japan), and the resulting homogenate, excluding the carapace, was collected as soft tissue. Lactate and ATP concentrations were measured after pretreatment using commercial assay kits (BM-LAC-100 and BM-ATP-100; Biomax, Guri, Republic of Korea), following the manufacturer’s protocols. The sampled tissue was centrifuged at 10,000× *g* for 10 min, and the supernatant was collected. For lactate measurement, 50 µL of solution was prepared by mixing 46 µL of supernatant with 2 µL of lactate enzyme mix and 2 µL of probe, then loaded into a 96-well plate. For ATP measurement, 50 µL of solution was prepared by mixing 44 µL of supernatant with 2 µL of ATP enzyme mix, 2 µL of probe, and 2 µL of ATP converter, and then added to a 96-well plate. All reaction mixtures were incubated for 30 min at room temperature along with the standard solutions provided in each kit. After incubation, absorbance at 570 nm was measured using a microplate reader (Multiskan SkyHigh, Thermo Fisher Scientific, Waltham, MA, USA). All absorbance data were converted into concentrations using a standard curve generated from the absorbance of the standard solutions.

### 2.4. Gene Expression of HSPs

To investigate the effect of anthropogenic vibrational disturbance on heat shock protein (HSP) expression, RNA isolation and cDNA synthesis were conducted using crab hepatopancreas samples before performing quantitative real-time PCR. The hepatopancreas was chosen as the tissue for RNA extraction because RNA concentrations appeared lower in the heart, gill, and eyestalk tissues (based on personal observation). All pre-treatments before real-time PCR were carried out according to the manufacturer’s protocols of the RNeasy^®^ Plus Mini Kit (Qiagen, Hilden, Germany) and the PrimeScript™ 1st Strand cDNA Synthesis Kit (Takara Bio, Kusatsu, Japan). Firstly, for RNA isolation from crab tissue, a 30 mg hepatopancreas sample was transferred into a 1.5 mL tube containing RLT buffer, and the supernatant was collected after centrifugation. The supernatant was then passed through a gDNA eliminator spin column via centrifugation. After removing the gDNA eliminator spin column, the flowthrough was mixed with 70% ethanol and loaded onto a RNeasy spin column. The sample was then washed by centrifugation using RW1 and RPE buffers, followed by RNA elution with RNase-free water. The concentration of the eluted RNA was measured using a uDrop Duo Plate (Thermo Fisher Scientific, Waltham, MA, USA) and a microplate reader. Secondly, for cDNA synthesis, 7 µL of RNA sample, diluted with DW to contain 1 µg based on its concentration, was mixed with a dNTP mixture, Oligo(dT)18 primer, and a random primer. The mixture was then incubated at 65 °C for 5 min. Following this, 5× 1st strand synthesis buffer, RNase inhibitor, PrimeScript RTase, and DW were added to the incubated solution, and the mixture was incubated at 42 °C for 1 h. After incubation, an additional incubation at 70 °C for 15 min was performed for enzyme inactivation. The final solution was stored at −80 °C for real-time PCR.

The primer for amplifying glyceraldehyde-3-phosphate dehydrogenase (GAPDH), used as a housekeeping gene, was designed based on the GAPDH sequence of *A. lactea* (GenBank accession No. KJ133056.1) (Table 1). The specific primers for HSP70 in *A. lactea* were designed by referencing primers previously used to amplify HSP70 in other ocypodid species, such as ghost crabs and other fiddler crabs. Ultimately, a PCR product was successfully amplified using the HSP70 primer from *Ocypode quadrata* (GenBank accession No. KU613078.1; primer sequence from [38]). The consensus sequence of the amplified product was retrieved using BLASTX (NCBI), and it showed a strong match with the HSP70 sequence of *Gammarus monticellus* (GenBank accession No. AHX41286.1), with an E-value of 1 × 10^−29^. The amino acid sequence translated from the consensus nucleotide sequence was analyzed using SnapGene Viewer (snapgene.com accessed on 2 April 2025). An overlapping region of amino acid sequences was identified between the PCR product and the HSP70 protein of *G. monticellus*. Based on the nucleotide sequence corresponding to this overlapping amino acid region, a new primer set for amplifying HSP70 in *A. lactea* was designed (Table 1). Thermal cycling for real-time PCR was carried out as follows: pre-denaturation at 95  °C for 2 min; 40 cycles of amplification at 95  °C for 30 s, 60  °C for 15 s, and 72  °C for 25 s; followed by a melt curve analysis from 65  °C to 95  °C at an increment of 0.05  °C per second. Meanwhile, the nucleotide sequence of the PCR product obtained in this study was aligned using BioEdit, and identity comparison with other fiddler crab species was conducted using Clustal Omega 1.2.4.

### 2.5. Statistical Analysis

All data were compared between the vibration exposure group and the control group. If the measured values were normally distributed (*p* > 0.05 in the Shapiro–Wilk test), an independent *t*-test was performed; otherwise, a Mann–Whitney *U* test was used. Additionally, Levene’s test was applied to test for heteroscedasticity. These statistical analyses were conducted using SPSS version 19.0 and R (R core [39]).

## 3. Results

### 3.1. Lactate and ATP Concentration

When exposes to vibration at approximately 100 dB re 1 µm/s^2^ at 120 Hz, the lactate concentration in the leg muscle of *Austruca lactea* was significantly lower than that of the control group (Mann–Whitney *U* test; *U* = 9, *n*_1_ = 7, *n*_2_ = 8, *p* = 0.043; Figure 3). However, the ATP concentration in the leg muscle did not differ significantly between the group exposed 120 Hz vibration and the control group (*U* = 21, *n*_1_ = 8, *n*_2_ = 7, *p* = 0.209; Figure 3). In contrast, under exposure to 250 Hz vibration, the lactate concentration in the leg muscle was significantly higher than that of the control group (*U* = 15, *n*_1,_ = *n*_2_ = 8, *p* = 0.043; Figure 3). The ATP concentration of crabs exposed to 240 Hz vibration was significantly lower than that of the controls (independent *t*-test; *t* = 2.41, *df* = 14, *p* = 0.015; Figure 3).

### 3.2. Gene Expression of Heat Shock Protein 70 kDa Gene Expression

To characterize the heat shock protein 70 kDa (HSP70) gene of *A. lactea*, a newly designed primer was used to amplify cDNA from *A. lactea*. A 161 bp nucleotide sequence was obtained (GenBank accession No. PV454699), which shared over 91% identity with that of other fiddler crab species (Table 2).

The HSP70 gene expression in the hepatopancreas of *A. lactea* exposed to 120 Hz vibration was not significantly different from that of the control group (independent *t*-test; *t* = 0.148, *df* = 6, *p* = 0.444; Figure 4). Under exposure to 250 Hz vibration, there was no significant difference in HSP70 gene expression between the exposure group and the control group (Mann–Whitney *U* test; *U* = 10, *n* = 4, *p* = 0.289; Figure 4). However, the variation in HSP70 gene expression among individuals was significantly higher in the 250 Hz exposure group than in the control group (Levene’s test; *F* = 6.999, *p* = 0.038).

## 4. Discussion

In this study, exposure to anthropogenic vibration at approximately 100 dB re 1 µm/s^2^ at the frequency of 250 Hz resulted in a higher lactate concentration and a lower ATP concentration in the leg muscle of *Austruca lactea* compared to the control group. Since Hwang et al. [40] reported increased movement velocity under the frequency of 250 Hz vibration, these physiological changes in muscle tissue may be related to the crabs’ locomotion. Full and Herrid [30] reported an increase in whole-body lactate following running in the fiddler crab (*Uca pugilator*), with greater lactate accumulation observed at higher velocities. Such increases in lactate during or after movement have been previously reported in crustaceans and are considered indicative of anaerobic metabolism [24,41,42,43]. Notably, the white leg shrimp (*Litopenaeus vannamei*) exhibited a fivefold increase in hemolymph lactate following a vigorous escape response compared to a resting group, and elevated lactate levels were maintained even after one hour of recovery [41]. These results suggest that the observed increase in lactate may have been induced by increased locomotion in response to vibrational disturbance. Meanwhile, it is crucial to replenish ATP to maintain adenylic energy charge after significant energy consumption required for cellular homeostasis, because vigorous escape responses can consume large amounts of energy [41]. In a study on red crabs (*Gecarcoidea natalis*), ATP levels were maintained for 45 min through compensatory increases in adenosine diphosphate (ADP) and adenosine monophosphate (AMP) [44]. Additionally, Full and Herrid [30] reported that more than 60% of the total ATP was generated through lactate production during vigorous exercise in the fiddler crab, *U. pugilator*. However, the ATP level in *A. lactea* decreased under a 250 Hz vibrational disturbance. These results suggest that anaerobic metabolism accompanied by increased lactate production may not have provided sufficient energy in this study. Furthermore, ATP production may have declined due to hemolymph acidification caused by production of lactate [45].

From the perspective of energy budget, such disruption of energy homeostasis could have a negative effect on *A. lactea*. In other words, anthropogenic vibrational disturbance could disrupt the strategies to maximize fitness by reducing cost-effectiveness of energy use. Since the priority of behavior can be altered based on benefits to fitness, it is important to determine the cost of a given behavior [46,47]. Additionally, male animals are required to strategically decide, within a given energetic budget, whether to consume more energy and time for courtship toward females, or to forage to obtain energy for courtship and survival [47]. In particular, in male fiddler crab courtship, the waving their large claws comprising approximately 50% of their body weight, involves substantial energetic cost [48,49]. Matsumasa and Murai [46] reported that vigorous waving for mating or agonistic behavior consumed more energy than cleaning and feeding, as indicated by increased lactate levels. Therefore, such energetic cost caused by anthropogenic vibrational disturbance may have a negative effect on the energy budget for courtship in male fiddler crabs. Furthermore, because vibrational disturbance can interrupt normal behaviors, particularly feeding [20], energy shortages may be aggravated. Meanwhile, there was no alteration in lactate and ATP concentrations under exposure to 120 Hz vibration. This could indicate that there was no energetic cost induced by 120 Hz vibration, in contrast to the behavioral response reported in Joo and Kim [20]. However, actual vibrational disturbances produced by human activities such as pile-driving consist of a broadband of frequencies, with particularly high energy radiated in the frequency band below 1000 Hz during pile-driving operations [35,36,37]. Additionally, the vibrations may vary depending on the type of substrate and environmental conditions [10,11]. Therefore, the potential negative effects of exposure to 120 Hz vibration on *A. lactea* should not be overlooked, as the behavioral responses observed under 120 Hz vibration exposure indicate the necessity of conducting field research during construction activities.

Under exposure to vibrations at both the frequencies of 120 Hz and 250 Hz, there was no significant difference in HSP70 gene expression in the hepatopancreas of *A. lactea* between the exposure group and the control group. However, this result does not indicate the absence of stress caused by anthropogenic vibrational disturbance, due to potential intraspecific variation. Although no remarkable increase in expression was observed in the frequency of 120 Hz exposure group, one individual in the frequency of 250 Hz exposure group showed noticeably the highest level of expression, resulting in significantly greater variation compared to the control group. The relative expression in this individual increased approximately 17 times higher than the control group. It is possible that the vibrational disturbance induced physiological stress. Further studies with more replicates are needed, as the results suggest potential intraspecific variation. HSPs are known to be sensitive indicators of environmental stress and play important roles in maintaining health. Various stressors have been shown to induce the synthesis of HSPs [25,50]. HSPs are classified according to molecular weight, sequence homology, and function. In particular, the upregulation of HSP27 and HSP70 has been reported in *Palinurus elephas* under exposure to underwater noise [25,51,52]. HSP70, in particular, is considered important for maintaining homeostasis and cellular viability [53]. Therefore, the marked increase in HSP70 expression observed in a fiddler crab in this study may indicate a physiological stress response to exposure to 250 Hz vibration, despite the absence of significant differences between the exposure and control groups.

This study provides physiological evidence that may be interpreted as a stress response to vibrational disturbance in the semi-terrestrial crab *A. lactea*. These indices could serve as biomarkers for investigating the effects of noise pollution, including vibrational and acoustic disturbances, on marine animals. In particular, this study aimed to link the behavioral responses reported in Joo and Kim [20] with the physiological responses observed here, to enhance understanding of the effects of vibrational disturbance [24]. The integration of multiple biomarkers can improve the comprehensiveness of such assessments [54]. Moreover, well-established biomarkers can facilitate the prediction of stressor impacts by serving as tools for early warning, exposure quantification, and effect detection [54]. In this study, biochemical analyses of muscle tissue may support the interpretation of behavioral responses. Furthermore, this study contributes useful molecular information for *A. lactea*, a species known to serve as a biomarker for environmental stress. Nonetheless, further integration of evidence is needed to improve the assessment of vibration and noise pollution effects on marine animals. For example, additional biochemical indices such as glucose levels and oxygen consumption could enhance evaluations of energy budgets [30,46,49]. The findings on HSP70 expression in this study also highlight the need for more rigorous investigation to validate its use as a biomarker [55].

## 5. Conclusions

In this study, the white-clawed fiddler crab (*Austruca lactea*) was exposed to anthropogenic vibrational disturbances at 120 and 250 Hz (~100 dB re 1 µm/s^2^) to investigate physiological responses. Under exposure to vibration at 250 Hz, lactate concentration increased in the leg muscle tissue, while ATP concentration decreased. It suggests a potential negative impact on the energy budget of *A. lactea* due to increased energetic cost. Additionally, the nucleotide sequence of HSP70—an indicator commonly used to assess environmental stress—was presented, along with newly designed primers for its amplification. A possible increase in HSP70 gene expression was observed, with variation among individuals, indicating potential intraspecific differences in response. These results suggest that anthropogenic vibrational disturbances may induce physiological stress in *A. lactea*. Furthermore, these physiological indicators could serve as biomarkers for assessing vibroacoustic pollution in marine animals. Further integration of various indicators, including behavioral and additional physiological responses, is needed to enhance the assessment of vibroacoustic pollution.

## Figures and Tables

**Figure 1 biology-14-00962-f001:**
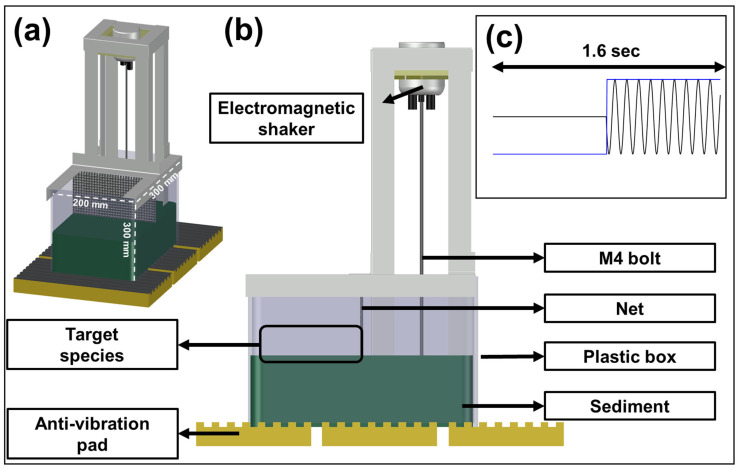
(**a**) 3D diagram of the experimental system used for the vibration exposure experiment. (**b**) Cross-sectional view of (**a**), with details provided in each text box. (**c**) Setup for signal generation using the function generator.

**Figure 2 biology-14-00962-f002:**
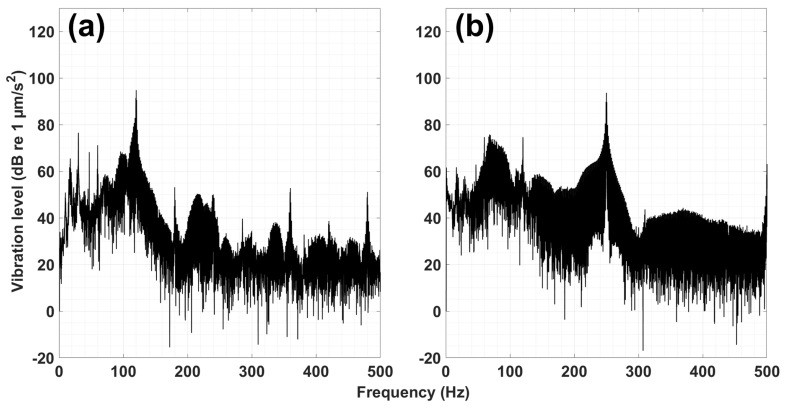
The spectra analyzed by FFT of (**a**) 120 and (**b**) 250 Hz vibrations.

**Figure 3 biology-14-00962-f003:**
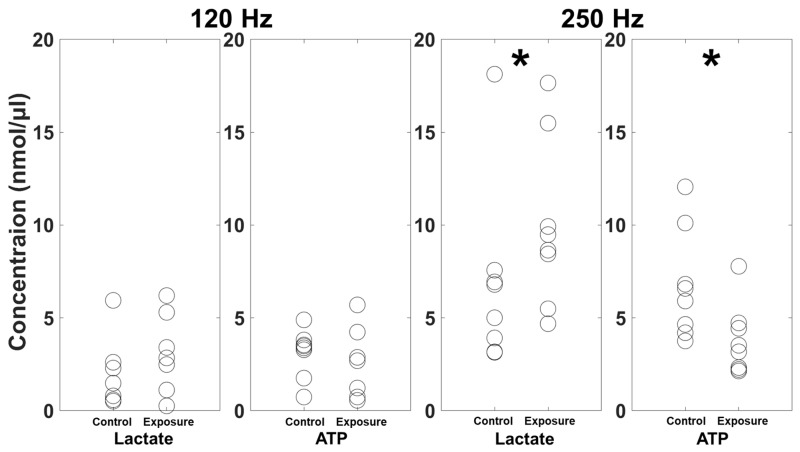
The lactate and ATP concentrations in the leg muscle of *Austruca lactea*. Asterisk (*) indicates statistical significance (*p* < 0.05).

**Figure 4 biology-14-00962-f004:**
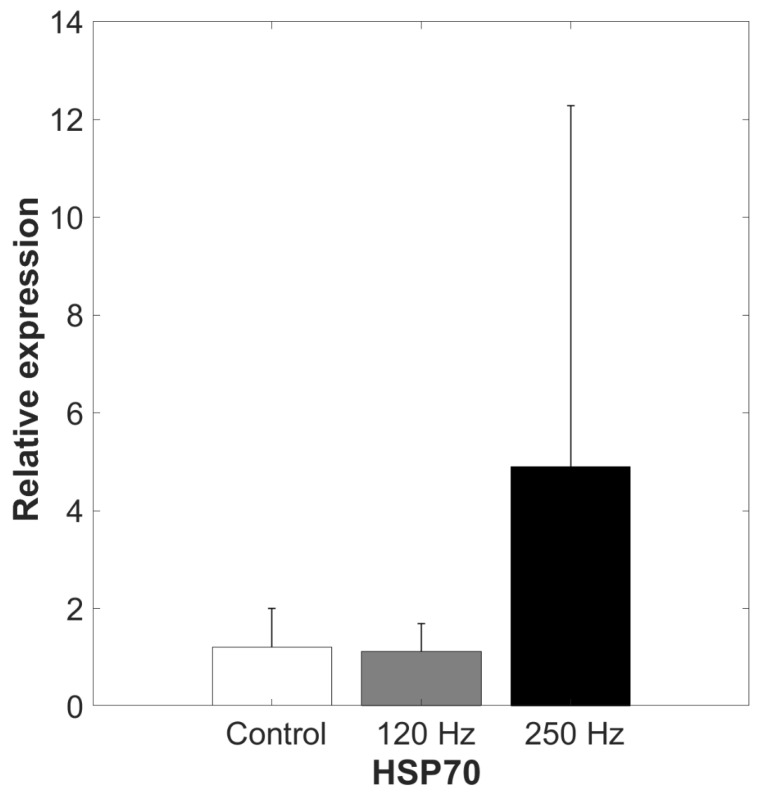
Relative gene expression of HSP70 in the hepatopancreas of *Austruca lactea* (mean ± standard error [SE]).

**Table 1 biology-14-00962-t001:** The Primer used for Real-Time PCR.

Primer Name	Sequence	GenBank Accession NO.
GAPDH (F)	CTTCTTGCACCACCAACTGC	KJ133056.1
GAPDH (R)	TCCACAACGGACACATCAGG	
HSP70 (F)	TGTACCGGCCTACTTCAACG	This study
HSP70(R)	AAGATGAGCACGTTGCGCT	

**Table 2 biology-14-00962-t002:** Percent Identity of the HSP70 Nucleotide Sequence of *Austruca lactea*. Compared with Other Fiddler Crab Species.

Species	Shared Identity (%)	GenBank Accession NO.
*Minuca mordax*	93.79	KC355776.1
*Minuca burgersi*	93.79	KF153223.1
*Minuca victoriana*	93.17	KC355779.1
*Minuca rapax*	93.17	KC355777.1
*Leptuca cumulanta*	93.17	KF153224.1
*Leptuca thayeri*	91.93	KC355778.1
*Leptuca uruguayensis*	91.93	KF153227.1
*Leptuca leptodactyla*	91.93	KF153225.1

## Data Availability

The raw data supporting the conclusions of this article will be made available by the authors on reasonable request.

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
