# Peer review of "The Physiological Response of the Fiddler Crab Austruca lactea to Anthropogenic Low-Frequency Substrate-Borne Vibrations"

_biology, 2025, doi:10.3390/biology14080962_

Round 1
Reviewer 1 Report
Comments and Suggestions for Authors
Dear authors, I really enjoyed reading this paper which details the stress response of a crustacean to substrate-borne vibration. It is a valuable contribution to the literature base. I have some comments which I suggest you think about:
Title and throughout- I would propose a slight change here, how about removing ‘anthropogenic’ and replacing with ‘low frequency’. It was not an anthropogenic source, it was a carefully controlled laboratory exposure to particular frequencies. Consider this phrasing here and throughout, although of course the link to anthropogenic sources must remain strong.
Abstract- lines 20 and 21 can reduce the method details (e.g. remove ‘using microplate reader and…’ etc
L38- consider citing ‘Roberts and Howard (2022) sub-borne vibrational noise in the Anthropocene’, as well
L38- Here, or later: I think you should go beyond this statement, the importance of vibrational noise is really taking off now and is being emphasised in recent assessments of anthropogenic sources. See ‘Sound-related effects of offshore wind energy on fishes and aquatic invertebrates- Williams et al., 2023’ and also ‘Marine energy converters: potential acoustic effects on fishes and aquatic invertebrates 2023’, and ‘Offshore wind energy development: research priorities..Popper et al JASA 2022’ …etc All of these papers have sections particularly regarding sub-borne stimuli. We also know, from the Roberts and Hazelwood papers (that you’ve cited elsewhere), that crustaceans can detect vibrational noise at the levels anthropogenic operations are producing them.
L44- add ‘Sholte waves’ (marine ‘counterpart’ to Rayleigh waves)
L55- 56- you could cite Radford and Stanley recent review paper on crustacean receptors here.
L55- around here: consider citing more of the older literature which particularly focusses on fiddler crab vibrational reception e.g. Salmon et al. 1977, Salmon 1971, Salmon and Atsaides 1969, Aicher and Tautz 1990, and also Budelmann ‘Hearing in crustacea’ 1992..
L67- here or elsewhere, you focus on fiddler crabs here but there have been vibrational noise studies on other crustaceans worth mentioning e.g. in field (Roberts and Laidre, 2019 ‘Finding a home’; Roberts et al 2017 'exposure of benthic'.. and other pubs from that author), but also very recent papers from the Aran Mooney lab at Woods hole (e.g. scallop paper measured sub-borne vibration, but check for more)
L118- Roberts et al. 2015 and Roberts et al 2016 could be cited here.
L123- is there justification for why you used only 120 Hz and 250 Hz anywhere? if not, that should be somewhere
L125- justification for this particular amplitude is required, is that realistic to a particular distance from a particular anthropogenic operation? This amplitude is certainly within the detection capabilities of most crustaceans however, so seems reasonable e.g. cite Roberts et al 2016 which has a summary figure of crustacean sensitivities.
L130- was the accelerometer attached to anything, or buried in the sediment? Where was the accelerometer in relation to the crab? Vibration will vary across the sediment so hopefully you measured the vibration next to the crab? some explanation needed.
L130- was this a single or tri-axial accelerometer? Worth mentioning which plane of motion the vibrational stimulus was greatest in.
L152- there are a low numbers of replicates in this study (N=8 per frequency group). I think that needs to be acknowledged in both the results and the discussion, as you have low statistical power.
Figure 2 – what were the background levels of vibration?
Figure 2- looking at the supplemental figures, there are harmonics for both 120 and 250 Hz. Probably worth mentioning these, and it would be interesting to know how much (e.g. what proportion) of the signal is at the desired frequencies and how much is not. For example in sensitivity experiments we aim for 95% of the signal to be at the desired frequency. This study is a noise exposure, so we can be less picky, but these exposures are certainly not just the desired frequencies, and I would be tempted to acknowledge that in methods.
L258- typo (double full-stop)
L267- was increased locomotion observed in this study>
L297- actual operations also occur over much longer periods of time, with vibrations highly variable according to the substrate type and environmental conditions at the time
Reference list- check all are cited in the text, I could not find the citation for Wale et al.
L304- here and elsewhere, be careful with phrasing, it is not only the frequency important here but the amplitude. At this particular frequency, at this particular amplitude, you have observed these reuslts. However emphasise the variability of freq/amplitude per source, within anthropogenic source etc
Reference list- check all references are cited in the text, I could not find the citation for Wale et al. (but perhaps I missed it!)
Author Response
Dear authors, I really enjoyed reading this paper which details the stress response of a crustacean to substrate-borne vibration. It is a valuable contribution to the literature base. I have some comments which I suggest you think about:
Comments 1: Title and throughout- I would propose a slight change here, how about removing ‘anthropogenic’ and replacing with ‘low frequency’. It was not an anthropogenic source, it was a carefully controlled laboratory exposure to particular frequencies. Consider this phrasing here and throughout, although of course the link to anthropogenic sources must remain strong.
Response 1: We added "low-frequency" to the title as suggested but kept "anthropogenic" (lines 1-2) because the vibration used in this study mimics pile-driving activity. Furthermore, the frequency considered in this study—particularly 250 Hz—is known to be a dominant frequency in the vibrational communication channel of Austruca lactea. Removing the term "anthropogenic" could lead to misunderstanding, as it may imply that "low-frequency" includes biologically generated signals such as those produced by animals.
Comments 2: Abstract- lines 20 and 21 can reduce the method details (e.g. remove ‘using microplate reader and…’ etc
Response 2: We deleted it as suggested (line 25).
Comments 3: L38- consider citing ‘Roberts and Howard (2022) sub-borne vibrational noise in the Anthropocene’, as well
Response 3: We added it as suggested (lines 41-42).
Comments 4: L38- Here, or later: I think you should go beyond this statement, the importance of vibrational noise is really taking off now and is being emphasised in recent assessments of anthropogenic sources. See ‘Sound-related effects of offshore wind energy on fishes and aquatic invertebrates- Williams et al., 2023’ and also ‘Marine energy converters: potential acoustic effects on fishes and aquatic invertebrates 2023’, and ‘Offshore wind energy development: research priorities..Popper et al JASA 2022’ …etc All of these papers have sections particularly regarding sub-borne stimuli. We also know, from the Roberts and Hazelwood papers (that you’ve cited elsewhere), that crustaceans can detect vibrational noise at the levels anthropogenic operations are producing them.
Response 4: We added it as suggested (lines 43-45).
Comments 5: L44- add ‘Sholte waves’ (marine ‘counterpart’ to Rayleigh waves)
Response 5: We added it as suggested (line 49).
Comments 6: L55- 56- you could cite Radford and Stanley recent review paper on crustacean receptors here.
Response 6: We added it as suggested (line 61).
Comments 7: L55- around here: consider citing more of the older literature which particularly focusses on fiddler crab vibrational reception e.g. Salmon et al. 1977, Salmon 1971, Salmon and Atsaides 1969, Aicher and Tautz 1990, and also Budelmann ‘Hearing in crustacea’ 1992..
Response 7: We added it as suggested (line 64).
Comments 8: L67- here or elsewhere, you focus on fiddler crabs here but there have been vibrational noise studies on other crustaceans worth mentioning e.g. in field (Roberts and Laidre, 2019 ‘Finding a home’; Roberts et al 2017 'exposure of benthic'.. and other pubs from that author), but also very recent papers from the Aran Mooney lab at Woods hole (e.g. scallop paper measured sub-borne vibration, but check for more)
Response 8: We added sentences and citations related to crustacean as suggested (lines 80-82).
Comments 9: L118- Roberts et al. 2015 and Roberts et al 2016 could be cited here.
Response 9: We added it as suggested in line 132.
Comments 10: L123- is there justification for why you used only 120 Hz and 250 Hz anywhere? if not, that should be somewhere
Response 10: In a previous study, Austruca lactea responded behaviorally only to vibrations at 120 and 250 Hz. Therefore, we conducted the exposure experiments based on those findings, we added this in lines 137-138.
Comments 11: L125- justification for this particular amplitude is required, is that realistic to a particular distance from a particular anthropogenic operation? This amplitude is certainly within the detection capabilities of most crustaceans however, so seems reasonable e.g. cite Roberts et al 2016 which has a summary figure of crustacean sensitivities.
Response 11: We added sentences related to this in lines 140-142.
Comments 12: L130- was the accelerometer attached to anything, or buried in the sediment? Where was the accelerometer in relation to the crab? Vibration will vary across the sediment so hopefully you measured the vibration next to the crab? some explanation needed.
Response 12: The accelerometer was placed together with the crab on the horizontal plane of the sediment surface. This information has been added in lines 148–149.
Comments 13: L130- was this a single or tri-axial accelerometer? Worth mentioning which plane of motion the vibrational stimulus was greatest in.
Response 13: We added it as suggested in line 148.
Comments 14: L152- there are a low numbers of replicates in this study (N=8 per frequency group). I think that needs to be acknowledged in both the results and the discussion, as you have low statistical power.
Response 14: Although obtaining more replicates would have been preferable, the number was limited due to several constraints. Nevertheless, statistically significant results were obtained.
Comments 15: Figure 2 – what were the background levels of vibration?
Response 15: The background vibration level in the control group was similar to that of the treatment group, except for the absence of the target frequency. As the control and treatment groups were tested simultaneously, the control group was isolated from experimental vibration using sufficient distance and an anti-vibration pad. These details have been added in lines 158–159.
Comments 16: Figure 2- looking at the supplemental figures, there are harmonics for both 120 and 250 Hz. Probably worth mentioning these, and it would be interesting to know how much (e.g. what proportion) of the signal is at the desired frequencies and how much is not. For example in sensitivity experiments we aim for 95% of the signal to be at the desired frequency. This study is a noise exposure, so we can be less picky, but these exposures are certainly not just the desired frequencies, and I would be tempted to acknowledge that in methods.
Response 16: Yes. As the reviewer pointed out, we had also checked for harmonics in both frequency experiments. To prevent these harmonics, the experimental box (or tank) should have over the 5 m of length (based on max velocity of compressional wave in dried sediment); It was impossible. However, these harmonics were lower than desired frequencies (approximately 50 dB re 1um/s2), we considered that is to be negligible.
Comments 17: L258- typo (double full-stop)
Response 17: We deleted it as suggested.
Comments 18: L267- was increased locomotion observed in this study>
Response 18: Empirically, yes. However, this was not quantified due to the absence of a video recording system.
Comments 19: L297- actual operations also occur over much longer periods of time, with vibrations highly variable according to the substrate type and environmental conditions at the time
Response 19: We added this as suggested in lines 307-308.
Comments 20: Reference list- check all are cited in the text, I could not find the citation for Wale et al.
Response 20: It was cited in line 87.
Comments 21: L304- here and elsewhere, be careful with phrasing, it is not only the frequency important here but the amplitude. At this particular frequency, at this particular amplitude, you have observed these reuslts. However emphasise the variability of freq/amplitude per source, within anthropogenic source etc
Response 21: The vibration amplitude was maintained at 107.22 ± 0.62 dB re 1 µm/s² (mean ± standard deviation) across all trials. Therefore, we referred to the amplitude as 'approximately 100 dB re 1 µm/s²' in the first mention within each section.
Comments 22: Reference list- check all references are cited in the text, I could not find the citation for Wale et al. (but perhaps I missed it!)
Response 22: If reference [23] is being referred to, it was cited in line 87.
Reviewer 2 Report
Comments and Suggestions for Authors
Review for the paper “The physiological response to anthropogenic substrate-borne vibrations of the fiddler crab Austruca lactea” by Soobin Joo and co-authors submitted to “Biology”.
The authors of this research paper conducted an analysis of the physiological effects of anthropogenic substrate-borne vibrations on the white-clawed fiddler crab (Austruca lactea). In their experiments, the crabs were exposed to controlled vibrational frequencies and assessed for changes in energy metabolism and stress-related responses. They found that exposure to vibrations influenced the metabolic activity of the crabs, particularly at higher frequencies. While no significant changes in ATP and lactate levels were observed at the lower frequency, the higher frequency resulted in reduced ATP concentrations and increased lactate levels, suggesting that the crabs may experience physiological strain from higher-frequency vibrational disturbances. The results of this study may have important implications for understanding the impact of human-generated vibrational noise on marine animals.
Recommendations.
Introduction.
L 36. The authors should clarify how anthropogenic vibrations compare to other anthropogenic stress sources in terms of intensity and prevalence in coastal environments.
L 66. The authors referenced that A. lactea locomotion behaviors changed under vibration exposure, such as decreased movement duration and increased walking velocity. They should explain whether these behavioral changes have long-term ecological consequences for species survival.
L 69. The authors pointed out that A. lactea is an endangered species in Korea. They should report whether anthropogenic vibrational pollution is considered a major threat to this species, relative to other factors such as habitat loss, climate change, or pollution from chemicals.
Methods:
L 102. The authors should include the sampling period, including the season and year. They should also include a map of the study site. What sampling method was used?
L 103. The authors should provide important details about the experimental crabs, including their size range, maturity status, and sex.
L 105. The authors explained that the acclimation aquarium was designed to ensure moist sediment and optimal conditions for A. lactea. They should clarify how long the crabs were acclimated in the laboratory.
L 125. The authors reported that vibration amplitude was maintained at 107.22 ± 0.62 dB re 1 µm/s2 and selected frequencies were 120 Hz and 250 Hz. They should clarify how this amplitude and frequencies compare to real-world vibrational intensities caused by human activities such as pile-driving.
L 162. Could the homogenization process have introduced variability in the measurements due to the inclusion of non-soft tissue (e.g., chitin or exoskeleton fragments)?
Results.
L 251. Levene's test is a statistical procedure used to assess the presence of heteroscedasticity in data. This is an inadequate approach when comparing different groups. The authors are advised to review the results. According to the authors' report, it appears that they employed the analysis of variance (ANOVA) method in this context. However, this method was not described in the "Methods",
Discussion.
L 256. The authors stated that exposure to anthropogenic vibration at 250 Hz resulted in physiological changes in muscle tissue. They should clarify whether the magnitude of vibration exposure (e.g., amplitude or intensity) was comparable to real-world disturbances, as this could affect the generalizability of the results. Was the vibration exposure duration also consistent with natural scenarios?
L 284. The authors referred to the potential energy costs of vibrational disturbance in male fiddler crab courtship behavior. How many male crabs were used in the experiments?
L 303. The authors highlighted that vibrational frequencies below 1000 Hz are common during human activities like pile-driving. They should clarify whether the 250 Hz and 120 Hz frequencies tested are representative of the broadband vibrational disturbances typically encountered near construction sites.
L 311. The authors reported intraspecific variation in HSP70 expression, with one individual showing a 17-fold increase in the 250 Hz group. They should hypothesize why this particular individual's response deviated significantly from others.
References.
The authors should follow the instructions for authors to properly format the references.
Author Response
Review for the paper “The physiological response to anthropogenic substrate-borne vibrations of the fiddler crab Austruca lactea” by Soobin Joo and co-authors submitted to “Biology”.
The authors of this research paper conducted an analysis of the physiological effects of anthropogenic substrate-borne vibrations on the white-clawed fiddler crab (Austruca lactea). In their experiments, the crabs were exposed to controlled vibrational frequencies and assessed for changes in energy metabolism and stress-related responses. They found that exposure to vibrations influenced the metabolic activity of the crabs, particularly at higher frequencies. While no significant changes in ATP and lactate levels were observed at the lower frequency, the higher frequency resulted in reduced ATP concentrations and increased lactate levels, suggesting that the crabs may experience physiological strain from higher-frequency vibrational disturbances. The results of this study may have important implications for understanding the impact of human-generated vibrational noise on marine animals.
Recommendations.
Introduction.
Comments 1: L 36. The authors should clarify how anthropogenic vibrations compare to other anthropogenic stress sources in terms of intensity and prevalence in coastal environments.
Response 1: We added sentences related to this (lines 41-42).
Comments 2: L 66. The authors referenced that A. lactea locomotion behaviors changed under vibration exposure, such as decreased movement duration and increased walking velocity. They should explain whether these behavioral changes have long-term ecological consequences for species survival.
Response 2: We added sentences as suggested in lines 78-79.
Comments 3: L 69. The authors pointed out that A. lactea is an endangered species in Korea. They should report whether anthropogenic vibrational pollution is considered a major threat to this species, relative to other factors such as habitat loss, climate change, or pollution from chemicals.
Response 3: The aim of our study is not to compare vibrational disturbance with other factors; therefore, we do not claim that vibrational disturbance poses a greater threat than other stressors. Instead, we highlighted that fiddler crabs, including Austruca lactea, are known for their remarkable ability to detect vibrations, as reported in previous studies (lines 61-71). This underscores the relevance of assessing the effects of vibrational disturbance specifically on A. lactea.
Methods:
Comments 4: L 102. The authors should include the sampling period, including the season and year. They should also include a map of the study site. What sampling method was used?
Response 4: We added sentences for collecting information in the manuscript (lines 111-114) and the map of collecting area in the supplementary materials.
Comments 5: L 103. The authors should provide important details about the experimental crabs, including their size range, maturity status, and sex.
Response 5: We added it as suggested in line 114-116.
Comments 6: L 105. The authors explained that the acclimation aquarium was designed to ensure moist sediment and optimal conditions for A. lactea. They should clarify how long the crabs were acclimated in the laboratory.
Response 6: We added this as suggested (line 117).
Comments 7: L 125. The authors reported that vibration amplitude was maintained at 107.22 ± 0.62 dB re 1 µm/s2 and selected frequencies were 120 Hz and 250 Hz. They should clarify how this amplitude and frequencies compare to real-world vibrational intensities caused by human activities such as pile-driving.
Response 7: We added sentences related to this in lines 140-142.
Comments 8: L 162. Could the homogenization process have introduced variability in the measurements due to the inclusion of non-soft tissue (e.g., chitin or exoskeleton fragments)?
Response 8: Not exactly. Suprenant was only used after homogenization.
Results.
Comments 9: L 251. Levene's test is a statistical procedure used to assess the presence of heteroscedasticity in data. This is an inadequate approach when comparing different groups. The authors are advised to review the results. According to the authors' report, it appears that they employed the analysis of variance (ANOVA) method in this context. However, this method was not described in the "Methods",
Response 9: We would like to clarify that our use of Levene’s test was not intended to compare group means, nor was it used as a substitute for ANOVA. Rather, it was employed solely to assess the equality of variances between the experimental and control groups, following the observation of an unusually large value in the experimental group.
As stated in the Results section, the primary comparison between groups was conducted using the Mann–Whitney U test, a non-parametric method appropriate for small sample sizes and potential deviations from normality. The Levene’s test was applied subsequently to evaluate the assumption of homogeneity of variance and to determine whether the large value in the experimental group may have influenced variance structure across groups.
Discussion.
Comments 10: L 256. The authors stated that exposure to anthropogenic vibration at 250 Hz resulted in physiological changes in muscle tissue. They should clarify whether the magnitude of vibration exposure (e.g., amplitude or intensity) was comparable to real-world disturbances, as this could affect the generalizability of the results. Was the vibration exposure duration also consistent with natural scenarios?
Response 10: The vibration settings were designed to resemble those generated by pile-driving operations, as referenced in previous studies on their vibrational characteristics. This information has been added in lines 140–142 of the Methods section.
Comments 11: L 284. The authors referred to the potential energy costs of vibrational disturbance in male fiddler crab courtship behavior. How many male crabs were used in the experiments?
Response 11: This was mentioned in the manuscript in lines 163–164, indicating that a total of 32 crabs were tested (N = 8 per frequency group).
Comments 12: L 303. The authors highlighted that vibrational frequencies below 1000 Hz are common during human activities like pile-driving. They should clarify whether the 250 Hz and 120 Hz frequencies tested are representative of the broadband vibrational disturbances typically encountered near construction sites.
Response 12: This study referred to Joo and Kim (2024), who reported that vibrations at 120 and 250 Hz affected locomotion. Their experiments were conducted using frequencies below 1000 Hz, which are known to carry high energy during pile-driving activities. Based on this, we conducted exposure experiments at 120 and 250 Hz. Additionally, our results do not imply that these frequencies are representative in human activities; rather, the primary focus is on the behavioral response of the fiddler crab.
Comments 13: L 311. The authors reported intraspecific variation in HSP70 expression, with one individual showing a 17-fold increase in the 250 Hz group. They should hypothesize why this particular individual's response deviated significantly from others.
Response 13: We added sentences related to this in lines 320-322.
References.
Comments 14: The authors should follow the instructions for authors to properly format the references.
Response 14: The reference style was revised according to the guidelines, using EndNote and the reference style file provided by MDPI.